# Association of the abbreviated burn severity index with mortality in severely burned patients: A meta-analysis

**Guangrong Deng[1], Ling Tang[1], Qian Yang[1], Zhengyong Li[2]\***

**1** Department of Anesthesiology, West China Hospital, Sichuan University/West China School of Nursing Sichuan University, Chengdu, Sichuan, China, **2** Department of Plastic and Burn Surgery, West China Hospital, Sichuan University, Chengdu, China

\* lizydd@sina.com

## Abstract

### Background and purpose

The ability of the abbreviated burn severity index (ABSI) to predict death among patients with severe burns remains unclear. This meta-analysis aimed to identify the association between the ABSI and mortality in severely burned patients.

### Methods

The PubMed, EMBASE and Web of Science databases were searched up to September 15, 2024. The odds ratios (ORs) with 95% confidence intervals (CIs) were combined, and a subgroup analysis was conducted on the basis of age, ABSI grouping method and OR source. All the statistical analyses were performed with STATA version 15.0.

### Results

Sixteen studies with 4011 cases were included in the analysis. The pooled results demonstrated that an elevated ABSI was significantly related to an increased risk of mortality (OR = 1.72, 95% CI: 1.48–2.00; P < 0.001). In addition, subgroup analysis by age (adult: OR = 1.35, P < 0.001; child: OR = 68.40, P < 0.001), ABSI grouping method (dichotomous: OR = 16.14, P < 0.001; continuous: OR = 1.59, P < 0.001) and OR source (univariate: OR = 11.42, P = 0.015; multivariate: OR = 1.51, P < 0.001) yielded similar results.

### Conclusion

The ABSI serves as a reliable prognostic indicator in severely burned patients, and patients with an elevated ABSI are at increased risk of death.

## Introduction

Burn injuries are among the most severe forms of skin damage and are among the leading causes of trauma. In recent years, their incidence has increased worldwide [1,2]. In 2019, over 8 million burn cases were reported worldwide, and more than 110,000 patients died from

**Data availability statement:** All data used in this work have been presented in the Supplementary Data File.

**Funding:** This work was funded by the Key Projects of Science and Technology Department of Sichuan Province(2024YFFK0063).

**Competing interests:** The authors have declared that no competing interests exist.

severe burns [3]. Typically, burns are initially assessed according to total body surface area (TBSA) affected and the depth of skin damage. First-degree burns are the most superficial thermal injuries, as they affect only the epidermis. Second-degree burns are further divided into superficial partial burns and deep partial burns, which generally involve the epidermis and dermis but not deeper structures. Full-thickness burns, described as third-degree burns, affect the entire epidermis, dermis, and underlying tissues [4,5]. Clinically, severe burns are the most challenging and often require surgical intervention.

Assessing the prognosis of patients with severe burns typically requires the consideration of multiple factors. The most commonly used indicators are the total body surface area (TBSA) burned, burn depth and age [6]. Generally, larger burn areas (greater than 30%) and deeper burns, such as deep partial-thickness and full-thickness burns (third-degree burns), are associated with worse outcomes [6]. In addition, young children and elderly individuals, due to their thinner skin and weaker immune function, tend to have poorer outcomes when experiencing burns of the same severity [7,8]. If a patient has inhalation injuries, such as those caused by the inhalation of hot smoke or chemical gases, and airway damage ensues, the prognosis worsens, which results in a higher mortality rate [9]. Additionally, underlying conditions such as diabetes, heart disease, and renal failure can impair the body's ability to recover from burns, which leads to a poorer prognosis [10]. Severe burn patients are prone to infections, especially those with extensive burns, and infection is one of the leading causes of death in these patients [11]. Therefore, the prognosis of patients with severe burns is based on multiple factors. Some scoring tools reported in the literature, such as the Baux score (based on age and burn area) and the revised Baux score (based on age, burn area and inhalation injury), still consider relatively few factors, which limits their clinical application [12,13].

The abbreviated burn severity index (ABSI), which was introduced in 1982, is a prognostic indicator that is based on sex, age, inhalation injury, occurrence of third-degree burns and total burned area [14]. In recent decades, this tool has been investigated in several studies since 2015 and is believed to have some prognostic value in severely burned patients. However, the results of these studies varied, and the clinical value of the ABSI in predicting mortality in patients with severe burns still lacks high-level evidence. Therefore, for the first time, this meta-analysis aimed to further clarify the association between the ABSI and the risk of mortality in patients with severe burns.

## Materials and methods

The meta-analysis in this study was performed according to the Preferred Reporting Items for Systematic Reviews and Meta-Analyses 2020 [15].

### Ethics Statement

The authors are accountable for all aspects of the work in ensuring that questions related to the accuracy or integrity of any part of the work are appropriately investigated and resolved. All procedures and studies that involved human participants were performed in accordance with the ethical standards of the institutional and/or national research committees and with the 1964 Declaration of Helsinki and its later amendments or comparable ethical standards.

### Literature search

We searched the PubMed, EMBASE and Web of Science databases from database inception up to September 15, 2024, for the following key words: abbreviated burn severity index, ABSI, burn, mortality, survival, prognosis, and death. The detailed search strategy was as follows: (abbreviated burn severity index OR ABSI) AND burn AND (mortality OR survival OR

prognostic OR prognosis OR death). MeSH terms and free texts were used, and the references cited in the included studies were reviewed.

### Inclusion criteria

Studies that met the following criteria were included: 1) patients included had severe burns, which are defined as those requiring transfer to a designated burn center for treatment according to established guidelines [16,17]; 2) the ABSI was evaluated according to sex (female: score 1; male: score 0), age (0–20 years: score 1; 21–40 years: score 2; 41–60 years: score 3; 61–80 years: score 4; 81–100 years: score 5), inhalation injury (with: score 1; without: score 0), third-degree burns (with: score 1; without: score 0) and total burned area (1–10%: score 1; 11–20%: score 2; 21–30%: score 3; 31–40%: score 4; 40–50%: score 5; 50–60%: score 6; 60–70%: score 7; 70–80%: score 8; 80–90%: score 9; 90–100%: score 10), with scores ranging from 2–19; and 3) the association of the ABSI values with mortality was explored.

### Exclusion criteria

Studies that met the following criteria were excluded: 1) those with insufficient, duplicated or overlapping data; 2) publications that were letters, editorials, animal trials, case reports or reviews; and 3) low-quality studies with a Newcastle–Ottawa Scale score less than 6 [18]; 4) studies without the OR and 95% CI for pooled analysis.

### Data collection

We collected the following information from the included studies: the first author, publication year, country, sample size, age, ABSI grouping method (cutoff value or continuous), source of OR (univariate or multivariate analysis), NOS, OR and 95% CI. Furthermore, for missing data except for the OR and 95% CI, we presented them as "not reported (NR)".

### Methodological quality assessment

All included studies were cohort studies. Therefore, the NOS scoring tool was used to assess the quality of the included studies. Only studies with an NOS score > 5 were included [18].

Two investigators independently performed the literature search, selection, data extraction, and methodological quality assessment, and all disagreements were resolved by team discussion.

### Statistical analysis

All the statistical analyses were performed via STATA (version 15.0) software. Heterogeneity between studies was assessed by $I^2$ statistics and the Q test. When significant heterogeneity was observed ($I^2 > 50\%$ and/or $P < 0.1$), the random-effects model was used; otherwise, the fixed-effects model was applied. ORs and 95% CIs were combined to evaluate the relationship between the ABSI and mortality. A subgroup analysis stratified by age, ABSI grouping method and OR source was then conducted. Next, a sensitivity analysis was conducted to detect the sources of heterogeneity and to assess the stability of the overall results. In addition, Begg's funnel plot and Egger's test were performed to detect publication bias, and significant publication bias was defined as $P < 0.05$ [19,20]. If obvious publication bias was observed, the trim-and-fill method was applied to clarify potentially unpublished studies [21].

## Results

### Literature search and selection

As presented in Fig 1, 741 publications were identified from three databases (PubMed: n = 223; EMBASE: n = 260; Web of Science: n = 258), and 130 duplicates were removed. In all, 453

and 98 records were excluded after reviewing the titles and abstracts, respectively. Eventually, after reviewing the full texts, 44 studies were further excluded (S1 Table). Therefore, 16 studies were included in the meta-analysis [22–37].

## Basic characteristics of the included studies

These 16 studies were published between 2015 and 2024 and included a total of 4011 participants. Most studies focused on adult patients, while one study enrolled only children. In five of the included studies, patients were divided into different groups according to the cutoff values of the ABSI, and the ABSI was analyzed as a continuous variable in the other seven studies. In most studies, the ORs and 95% CIs were extracted from the multivariate analysis



**Fig 1. Prisma flow diagram of this study.**

**Table 1. Basic characteristics of included studies.**

| Author | Year | Country | Sample size | Age (year-old) | Cutoff value of ABSI | Source of OR | NOS |
|---|---|---|---|---|---|---|---|
| Heng [22] | 2015 | UK | 90 | 45.7 (31.8-58.8) | 6 | U | 6 |
| Pantet [23] | 2016 | Switzerland | 492 | 42 ± 30 | Continuous | U | 7 |
| Yoon [24] | 2017 | Republic of Korea | 84 | 52 ± 14.6 | 10 | M | 6 |
| Barcellos [25] | 2018 | Brazil | 140 | <16 | 7 | U | 6 |
| Shahi [26] | 2018 | Iran | 105 | 30.1 ± 12.1 | Continuous | M | 6 |
| Ding [27] | 2019 | China | 127 | 43 ± 9 | Continuous | M | 6 |
| Chen [28] | 2020 | China | 128 | 44.9 ± 16.8 | Continuous | M | 6 |
| Depret [29] | 2020 | France | 111 | 48 (32.5-63) | Continuous | M | 6 |
| Zeng [30] | 2020 | China | 231 | 18-60 | Continuous | M | 6 |
| Lin [31] | 2021 | China | 60 | 43.7 ± 7.3/43.6 ± 10.7 | Continuous | M | 6 |
| Lin [32] | 2022 | China | 590 | 45.8 ± 16.2 | Continuous | M | 7 |
| Tsolakidis [33] | 2022 | Germany | 252 | 39 (28-56) | Continuous | M | 6 |
| Jiang [34] | 2023 | China | 194 | 48.7 ± 12.4/48.5 ± 15.7 | 12 | U | 6 |
| Niculae [35] | 2023 | Romania | 93 | 55.80 ± 17.16 | 9 | U | 6 |
| Nitescu [36] | 2023 | Romania | 121 | 61.19 ± 16.79/50.47 ± 14.62 | Continuous | M | 6 |
| Christ [37] | 2024 | Austria | 1193 | NR | Continuous | M | 6 |

ABSI: abbreviated burn severity index; OR: odds ratio; NOS: Newcastle-Ottawa Scale; U: univariate; M: multivariate; NR: not reported.

(11/16). Other data are presented in Table 1. Besides, the detailed information for the NOS score was shown in S2 Table.

## Associations between the ABSI and mortality in patients with severe burns

The pooled results demonstrated that an elevated ABSI significantly increased the risk of mortality (OR = 1.72, 95% CI: 1.48–2.00, P < 0.001; I² = 95.6%, P < 0.001) (Fig 2).

A subgroup analysis on the basis of age (adult: OR = 1.35, 95% CI: 1.20–1.51, P < 0.001; child: OR = 68.40, 95% CI: 9.11–513.81, P < 0.001), ABSI grouping method (dichotomous: OR = 16.14, 95% CI: 3.83–68.03, P < 0.001; continuous: OR = 1.59, 95% CI: 1.38–1.84, P < 0.001) and source of OR (univariate: OR = 11.42, 95% CI: 1.61–81.17, P = 0.015; multivariate: OR = 1.51, 95% CI: 1.31–1.74, P < 0.001) yielded similar results (Table 2).

## Sensitivity analysis and publication bias

The sensitivity analysis revealed that our results were stable and reliable and that none of the included studies had a significant effect on the overall conclusion (Fig 3).

Begg's funnel plot (Fig 4) and Egger's test (P = 0.003) indicated significant bias. Therefore, the trim-and-fill method was applied, and seven potentially unpublished studies were detected (Fig 5). However, these seven studies did not affect the overall findings and had an OR of 1.48 (95% CI: 1.28–1.70, P < 0.001).

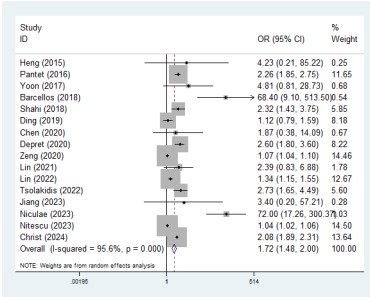

**Fig 2. Association of abbreviated burn severity index with mortality among severely burned patients.**

**Table 2. Results of meta-analysis.**

| Items | Number of studies | Odds ratio | 95% confidence interval | P value | I² (%) | P value |
|---|---|---|---|---|---|---|
| Overall | 16 | 1.72 | 1.48–2.00 | P < 0.001 | 95.6 | P < 0.001 |
| Age | | | | | | |
| Adult | 13 | 1.35 | 1.20–1.51 | P < 0.001 | 88.3 | P < 0.001 |
| Child | 1 | 68.40 | 9.11–513.81 | P < 0.001 | – | – |
| Grouping method | | | | | | |
| Dichotomous ABSI | 5 | 16.14 | 3.83–68.03 | P < 0.001 | 57.8 | 0.050 |
| Continuous ABSI | 11 | 1.59 | 1.38–1.84 | P < 0.001 | 96.5 | P < 0.001 |
| Source of OR | | | | | | |
| Univariate analysis | 5 | 11.42 | 1.61–81.17 | 0.015 | 87.8 | P < 0.001 |
| Multivariate analysis | 11 | 1.51 | 1.31–1.74 | P < 0.001 | 95.8 | P < 0.001 |

ABSI, abbreviated burn severity index; OR, odds ratio; U, univariate; M, multivariate.

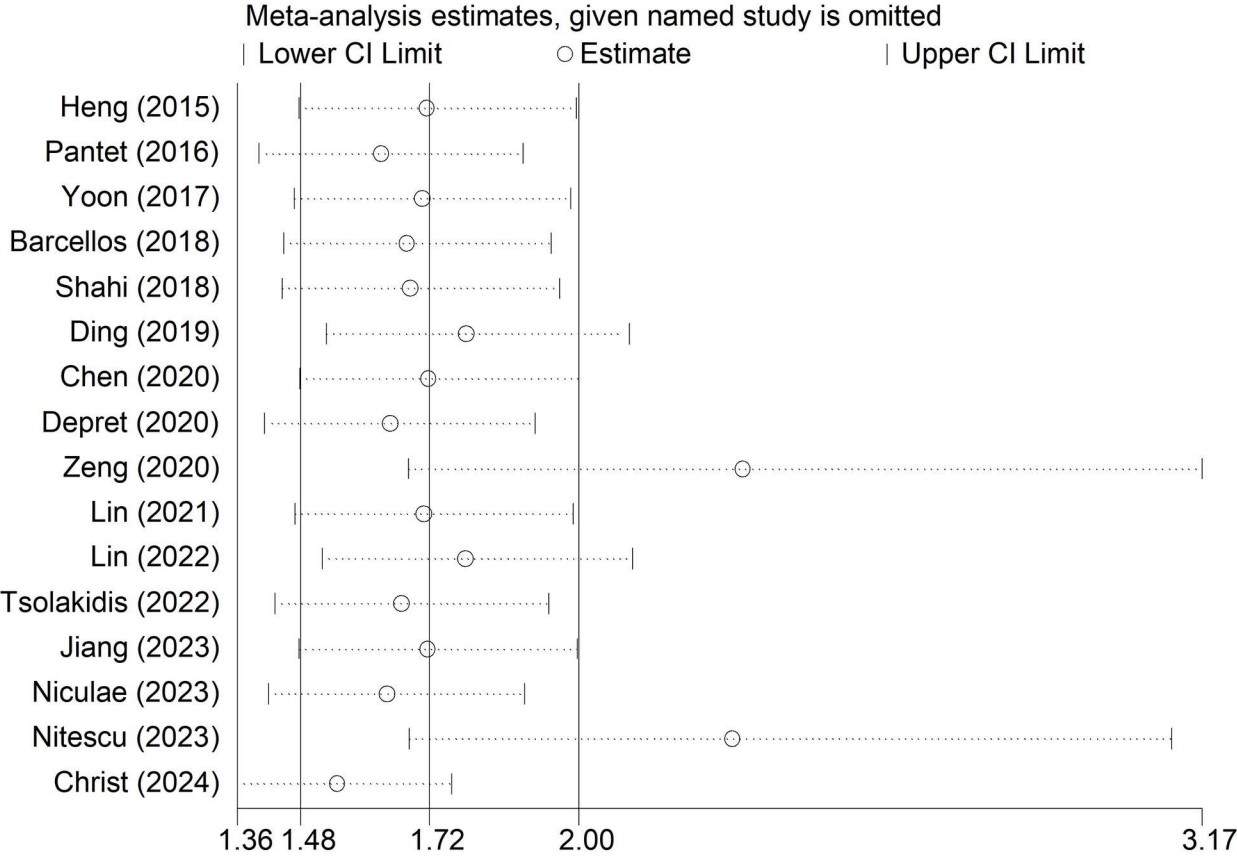

**Fig 3. Sensitivity analysis for the association of abbreviated burn severity index with mortality among severely burned patients.**

## Discussion

Our meta-analysis demonstrated that the ABSI is a reliable and valuable predictive indicator of mortality in severely burned patients and that patients with higher ABSI have an increased risk of mortality. Moreover, the subgroup analysis on the basis of age, ABSI grouping method and OR source further verified the above findings. Our study provides high-level evidence for the clinical application of the ABSI in severely burned patients.

The direct causes of death in patients with severe burns primarily include infection and multiple organ failure. Severe burns lead to destruction of the skin barrier, which increases the vulnerability of patients to bacterial, viral, and fungal infections. Consequently, post-burn infections, particularly sepsis, are among the main causes of mortality [38,39]. Additionally, extensive burns can trigger systemic inflammatory response syndrome (SIRS), which leads to the failure of multiple organs, such as the heart, lungs, and kidneys, ultimately resulting in death [40]. Furthermore, the inhalation of hot smoke, toxic gases, or chemicals can cause airway damage and result in acute respiratory distress syndrome (ARDS) or pneumonia, which worsens the patient's condition and increases mortality rates [41,42]. Fluid loss and increased vascular permeability after burns can easily lead to hypovolemic shock, and if timely resuscitation and fluid replacement are not provided, it can lead to organ ischemia and death. Severe burns can also cause electrolyte and metabolic disturbances, such as hyperkalemia, hyponatremia, and acidosis, which can affect the function of the heart and other organs, posing life-threatening risks [43]. Other factors, such as acute

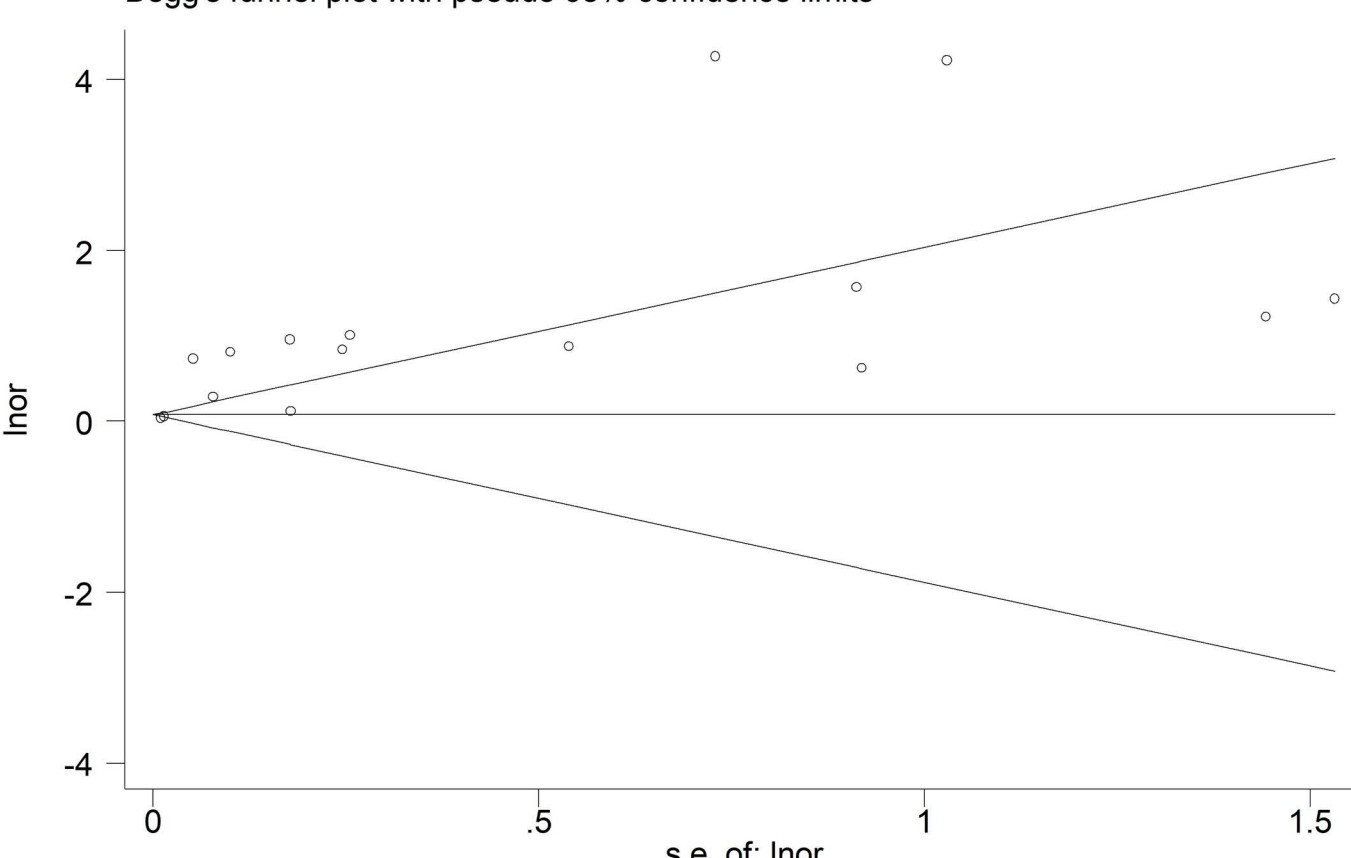

**Fig 4. Beggs' funnel plot for the association of abbreviated burn severity index with mortality among severely burned patients.**

renal failure, malnutrition, and decreased immunity, are also risk factors that can exacerbate this condition [44,45]. However, these factors often do not exist in isolation, and the death of burn patients is typically the result of multiple contributing factors. The indicators included in the ABSI, such as age, burn area, and the occurrence of third-degree burns, can effectively reflect the risk of developing the aforementioned complications, thereby predicting the risk of mortality.

The ABSI score, introduced in 1982, has been widely utilized as a prognostic tool for assessing mortality risk in burn patients [14]. However, over the past few decades, the validity of the ABSI score has been questioned because of advancements in burn care and the emergence of more sophisticated prognostic tools. One of the main criticisms is that the original score was developed based on patient data from a different era, with medical practices and outcomes that may not fully align with contemporary standards. For example, improvements in intensive care, fluid resuscitation, infection control, and surgical interventions have significantly enhanced patient survival, particularly among those with severe burns [46]. Additionally, some studies have suggested that the weight of certain variables in the ABSI score, such as sex or age, may not reflect the same level of prognostic significance in modern clinical settings [47]. Despite these criticisms, recent studies have reaffirmed the utility of the ABSI score, particularly in resource-limited settings or as a baseline comparison for newer predictive models [48]. These findings underscore the importance of continuously validating

## Filled funnel plot with pseudo 95% confidence limits

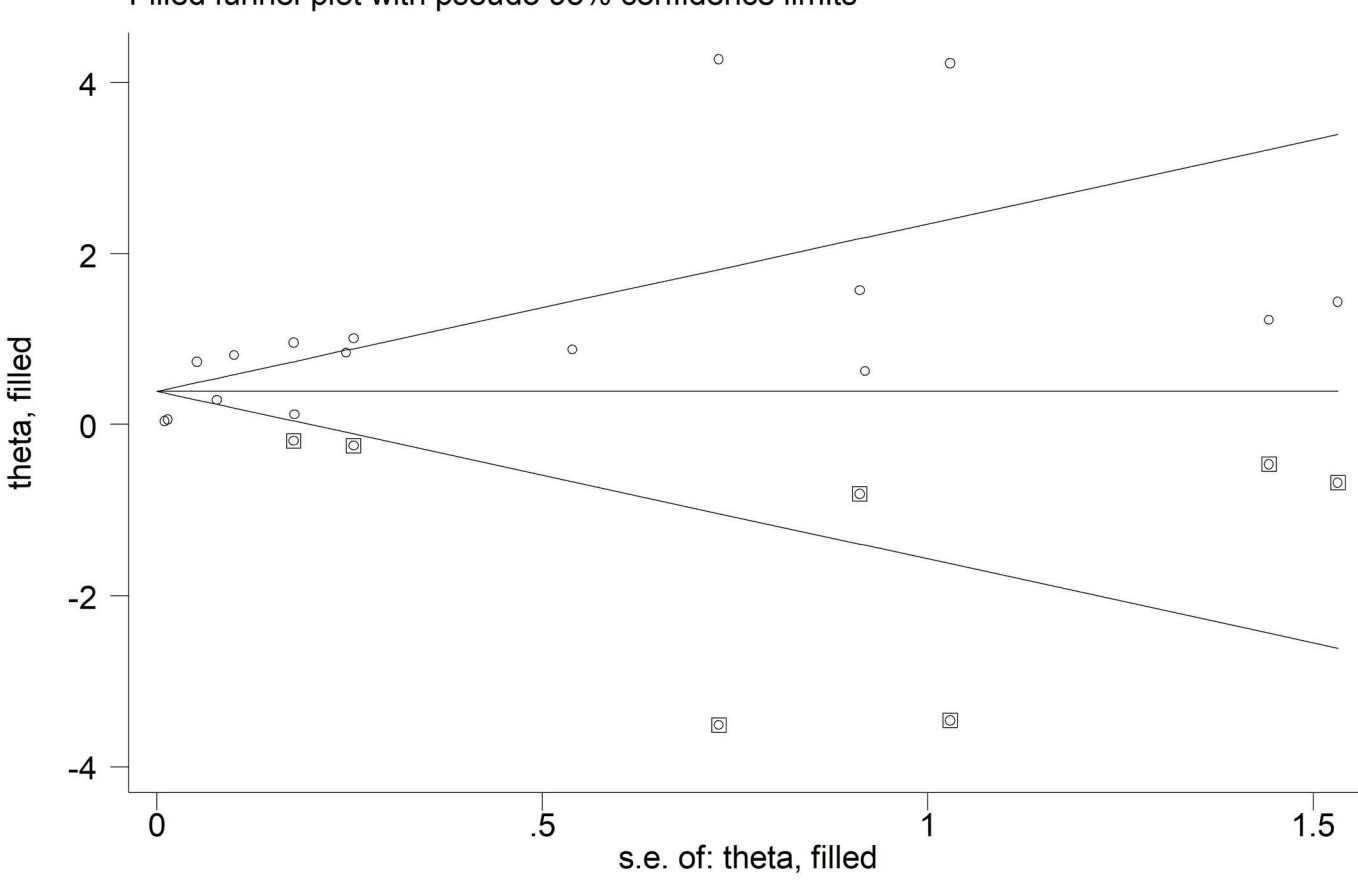

**Fig 5. Fille funnel plot for the association of abbreviated burn severity index with mortality among severely burned patients.**

and updating prognostic tools such as the ABSI to ensure their relevance in guiding clinical decision-making and improving patient outcomes.

Among the included studies, only one focused on pediatric patients [25]. However, as we mentioned in the inclusion criteria, age is an important parameter of the ABSI, and patient age ≤ 20 years is defined as a score of 1. Therefore, the prognostic value of the ABSI in children requires further validation or adjustment. In addition, our meta-analysis focused on the ability of the ABSI to predict the risk of death in patients with severe burns. The associations of ABSI with other postburn complications, such as infection, acute renal failure, arrhythmia and cicatrization, should also be explored, but relevant studies are still limited. In most of the included studies, the ABSI was treated as a continuous variable, whereas in some studies, it was categorized as a binary variable. Although we performed a subgroup analysis to address this issue, for better clinical application of the ABSI in patient risk assessment, more precise risk stratification may still be needed.

This meta-analysis has several limitations. First, the overall sample size was relatively small, and most included studies were retrospective, which might cause some bias. Second, the heterogeneity discovered during the meta-analysis was high, and identifying the source of high heterogeneity was difficult. Third, we were unable to conduct more subgroup analyses based on other important parameters, such as the type of burn and cutoff value of the ABSI, due to the lack of original data. Fourth, we did not compare the prognostic value of the ABSI

with that of other similar indicators, such as the Baux score, which is also difficult to achieve in meta-analyses because of the lack of original data.

## Conclusion

The ABSI serves as a reliable prognostic indicator in patients with severe burns. Unlike previous studies, this meta-analysis provides robust evidence for the clinical utility of the ABSI. Our findings reinforce that patients with elevated ABSI scores are at an increased risk of mortality, which emphasizes the importance of this indicator in risk stratification and in guiding treatment prioritization in modern burn care. However, due to the limitations of our meta-analysis and in the included studies, more high-quality studies are still needed to verify our findings.

## Suppporting information

**S1 Table. Reasons for inclusion or exclusion of all publications.**
(XLSX)

**S2 Table. Detailed results of NOS score.**
(DOCX)

**S1 File. All relevant data used in this work.**
(XLSX)

## Author contributions

**Conceptualization:** Guangrong Deng, Zhengyong Li.

**Data curation:** Ling Tang.

**Formal analysis:** Guangrong Deng, Ling Tang, Qian Yang.

**Investigation:** Ling Tang.

**Methodology:** Guangrong Deng, Qian Yang.

**Resources:** Qian Yang.

**Software:** Qian Yang.

**Supervision:** Zhengyong Li.

**Validation:** Ling Tang.

**Writing – original draft:** Guangrong Deng.

**Writing – review & editing:** Zhengyong Li.

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
