## [Decision Letter · Decision Letter 0]

17 Dec 2024

PONE-D-24-45843Association of abbreviated burn severity index with mortality in severely burned patients: a meta-analysisPLOS ONE

Dear Dr. Li,

Thank you for submitting your manuscript to PLOS ONE. After careful consideration, we feel that it has merit but does not fully meet PLOS ONE’s publication criteria as it currently stands. Therefore, we invite you to submit a revised version of the manuscript that addresses the points raised during the review process.

Thank you for your patience with the reviews of this manuscript.  In short, you will see a number of suggestions that are really needed to improve this story for acceptance. Particular attention to the comments of Reviewer 1 and Reviewer 2 are needed. Overall, more in depth analysis of the ASBI needs to be done, and what stands this work apart from previous studies on the ASBI is needed. Additionally, your conclusions need to be updated. Presumably, this score includes age, TBSA, and inhalation injury, as was developed over 4 decades ago. Your conclusion that "ABSI could serve as a novel and reliable prognostic indicator in severely burned patients" is not true because the ASBI is not novel. Was there any comparison to other scoring systems (e.g., Baux)? Moreover, your conclusion that "patients with an elevated ABSI are exposed to increased risk of death." is also not new- age and TBSA have been known predictors of mortality for nearly 70 years. As such, I encourage your new conclusions to better highlight the uniqueness of this work, and what it brings to the field.

We look forward to receiving your revised manuscript.

Kind regards,

David M. Burmeister, PhD

Academic Editor

PLOS ONE

Journal Requirements:

“This work was funded by the Key Projects of Science and Technology Department of Sichuan Province(2024YFFK0063).”

“This work was funded by the Key Projects of Science and Technology Department of Sichuan Province(2024YFFK0063).”

5. We note that your Data Availability Statement is currently as follows: [All data used in this work have been presented in the manuscript.]

6. As required by our policy on Data Availability, please ensure your manuscript or supplementary information includes the following:

Additional Editor Comments (if provided):

Dr. Li,

Thank you for your patience with the reviews of this manuscript.

In short, you will see a number of suggestions that are really needed to improve this story for acceptance. Particular attention to the comments of Reviewer 1 and Reviewer 2 are needed. Overall, more in depth analysis of the ASBI needs to be done, and what stands this work apart from previous studies on the ASBI is needed.

Presumably, this score includes age, TBSA, and inhalation injury, as was developed over 4 decades ago. Your conclusion that "ABSI could serve as a novel and reliable prognostic indicator in severely burned patients" is not true because the ASBI is not novel. Was there any comparison to other scoring systems (e.g., Baux)? Moreover, your conclusion that "patients with an elevated ABSI are exposed to increased risk of death." is also not new- age and TBSA have been known predictors of mortality for nearly 70 years.

As such, I encourage you to better highlight the uniqueness of this work, and what it brings to the field.

Reviewers' comments:

Reviewer's Responses to Questions

**Comments to the Author**

1. Is the manuscript technically sound, and do the data support the conclusions?

Reviewer #1: Yes

Reviewer #2: Partly

Reviewer #3: Yes

Reviewer #4: Yes

2. Has the statistical analysis been performed appropriately and rigorously? 

Reviewer #1: Yes

Reviewer #2: I Don't Know

Reviewer #3: Yes

Reviewer #4: Yes

3. Have the authors made all data underlying the findings in their manuscript fully available?

Reviewer #1: Yes

Reviewer #2: Yes

Reviewer #3: Yes

Reviewer #4: No

4. Is the manuscript presented in an intelligible fashion and written in standard English?

Reviewer #1: No

Reviewer #2: Yes

Reviewer #3: Yes

Reviewer #4: Yes

5. Review Comments to the Author

Reviewer #1: This manuscript analyzes the prognostic relevance of the ABSI score based on 16 studies published between 2015 and 2024. The statistical analysis is thorough and the results are presented in a clear and understandable form. However, the overall quality of the text lacks in multiple areas. Overall, many of the chosen formulations are inadequate and articles are often missing, which makes the text not very pleasant to read.

The authors claim, that the ABSI score could possibly be used as a novel prognostic score, which doesn't make any sense as the score was introduced in 1982 and has been in use ever since. It is true, that the relevance of the score has been questioned in the past, however, there are multiple (very recent) studies which test the score for its validity. However, the authors repeatedly claim in this manuscript that the ABSI score is novel, which could be very misleading for readers.

Furthermore, the authors don't provide enough references for the claims in the introduction as well as the discussion. Following claims need to be undermined with references:

Introduction:

- Young children and the elderly, due to thinner skin and weaker immune function, tend to have poorer

outcomes when faced with burns of the same severity. (??)

- Additionally, underlying conditions such as diabetes, heart disease, and renal failure can impair the

body's ability to recover from burns, leading to a poorer prognosis. (e.g. Resch A, Neumueller A, Christ A, Staud C, Hacker S. Chronic kidney disease and cardiovascular disease reduce survival rates after burn injury: A retrospective study over 20 years.)

- Severe burn patients are prone to infections, especially with extensive burns, and infection is one of the

leading causes of death in these cases.

- However, due to weaker immunity and lower treatment compliance, children may have a worse prognosis compared to adult patients in the same situation. (The ABSI score is especially relevant for severely burned patients, who require intensive care treatment. There is no such thing as "lower treatment compliance" in children in ICU's.)

In the materials and methods section the authors write "Studies met following criteria were included: 1) severely burned patients with the degree II burns ≥30% or degree II burns ≥10% of the total body surface area,...." which doesn't make any sense and needs to be rewritten and clarified.

Overall the discussion is weak and lacks in relevant information about the ABSI score and why it's validity has been questioned over the past decades.

I would highly suggest extensive major revision, as the findings of this meta-analysis are clinically relevant and statistic analysis seems thorough, however the manuscript lacks in multiple areas. The text must be revised by a native english speaker and above mentioned points have to be revisited by the authors.

Reviewer #2: ABSTRACT:

* The conclusion should be a statement in the affirmative rather than subjuctive, i.e. 'ABSI serves as a reliable prognostic indicator for mortality in burned patients'. By the way, it is not novel,... This phrase should be deleted.

INTRODUCTION:

* Age is included as a major factor in all useful predictors, which are burn size and age and should be introduced together.

* ABSI has been around for a long time, and the major inclusion over age and burn size alone (a.k.a. Baux score) is inhalation injury, which goes back and forth. The paper should really include an analysis of Baux score too. Is ABSI better than the Baux score in a meta-analysis?

METHODS:

* How exactly were the studies selected? Was inclusion confirmed by more than one independent assessor? Were PRISMA guidelines used?

RESULTS:

* What exactly does 'elevated ABSI' mean? The statistical methodology should be much clearer. This should probably be done with categorical variables. For instance, an OR of 68 is massive and not all that instructive. Further, most evidence would suggest that this is non-linear which is not really reflected in the analysis.

DISCUSSION:

* What is the range of ABSI? The amount of data available suggest a much more detailed analysis.

Reviewer #3: Dear author

Congratulations about this work

I suggest that you correct line 15 degree II to Degree III > 10%

And in discussion if you found another characteristics related to mortality beyond ABSI ( as face, inhalation) i suggest that you commented

thanks

Reviewer #4: The article entitled "Association of abbreviated burn severity index with mortality in severely burned patients: a meta-analysis" aimed to validate a commonly used tool in burn research to predict mortality, performed as a meta-analysis of data from around the world.

The aims of this study are important to both research and clinical practice. From a technical standpoint the meta-analysis is performed well.

Suggestions:

•Consider discussing the ABSI more – even showing the formula – so that the general readership understands how it works and how it is applied

•At minimum, the original paper that introduced the ABSI (Tobiasen J, Hiebert JM, Edlich RF. The abbreviated burn severity index. Ann Emerg Med. 1982;11:260–2.) should be referenced.

•You report odds ratios for the ABSI. As a clinician who would use this tool in practice, this is not a particularly useful statistic. I would consider elements such as sensitivity, specificity, PPV, NPV, and probably most helpful, ROC curves, to help communicate the accuracy of the ABSI. There are at least 10 tools to predict mortality in burns, all with pros and cons. The paper should ideally make this as clinically relevant as possible.

6. PLOS authors have the option to publish the peer review history of their article (what does this mean? ). If published, this will include your full peer review and any attached files.

**Do you want your identity to be public for this peer review?** For information about this choice, including consent withdrawal, please see our Privacy Policy .

Reviewer #1: No

Reviewer #2: No

Reviewer #3: No

Reviewer #4: No

---

## [Author Response · Author response to Decision Letter 0]

4 Jan 2025

Response to Editor: (Second round)

1. Thank you for responding to our latest query. Before we can proceed with your manuscript, please could you provide the following details:

1) In the PRISMA flowchart, you specify that originally 611 records were screened for this study. Please provide a numbered table of all those 611 studies identified in the literature search, including those that were excluded from the analyses. For every excluded study, the table should list the reason(s) for exclusion. If any of the included studies are unpublished, include a link (URL) to the primary source. An example for how we would expect a table like this to look like can be found here: https://journals.plos.org/plosone/article?id=10.1371/journal.pone.0313866#sec017 (S2 file)

Answer: We have updated the Supplementary table 1 to list all 741 publications and reasons for exclusion.

2) Please also update the methods section of your manuscript with an explanation of how missing data were handled.

Answer: In the exclusion, we added the sentence: “4) studies without the OR and 95% CI for pooled analysis.”. And also “Furthermore, for missing data except for the OR and 95% CI, we presented them as “not reported (NR)”.”

3) Please provide a table showing the completed risk of bias and quality/certainty assessments for each study or outcome.

Answer: We have added the supplementary table 2 to show the completed risk of bias and quality/certainty assessments for each study or outcome.

Response to Academic Editor:

Question 1: In short, you will see a number of suggestions that are really needed to improve this story for acceptance. Particular attention to the comments of Reviewer 1 and Reviewer 2 are needed. Overall, more in depth analysis of the ASBI needs to be done, and what stands this work apart from previous studies on the ASBI is needed.

Answer: Dear editor, many thanks for your valuable comments and nice work. We sincerely appreciate the opportunity to revise our manuscript. We have carefully modified our manuscript according to the comment by all reviewers. Besides, we also discussed this issue in the manuscript to highlight the necessity of this study as follows: “The ABSI score, introduced in 1982, has been widely utilized as a prognostic tool for assessing mortality risk in burn patients [14]. However, over the past few decades, the validity of the ABSI score has been questioned because of advancements in burn care and the emergence of more sophisticated prognostic tools. One of the main criticisms is that the original score was developed based on patient data from a different era, with medical practices and outcomes that may not fully align with contemporary standards. For example, improvements in intensive care, fluid resuscitation, infection control, and surgical interventions have significantly enhanced patient survival, particularly among those with severe burns [46]. Additionally, some studies have suggested that the weight of certain variables in the ABSI score, such as sex or age, may not reflect the same level of prognostic significance in modern clinical settings [47]. Despite these criticisms, recent studies have reaffirmed the utility of the ABSI score, particularly in resource-limited settings or as a baseline comparison for newer predictive models [48]. These findings underscore the importance of continuously validating and updating prognostic tools such as the ABSI to ensure their relevance in guiding clinical decision-making and improving patient outcomes.”

Question 2: Additionally, your conclusions need to be updated. Presumably, this score includes age, TBSA, and inhalation injury, as was developed over 4 decades ago. Your conclusion that "ABSI could serve as a novel and reliable prognostic indicator in severely burned patients" is not true because the ASBI is not novel. Was there any comparison to other scoring systems (e.g., Baux)? Moreover, your conclusion that "patients with an elevated ABSI are exposed to increased risk of death." is also not new- age and TBSA have been known predictors of mortality for nearly 70 years. As such, I encourage your new conclusions to better highlight the uniqueness of this work, and what it brings to the field.

Answer: Dear editor, thanks for your valuable question. We have carefully revised our conclusion as request. Conclusion in the abstract: “The ABSI serves as a reliable prognostic indicator in severely burned patients, and patients with an elevated ABSI are at increased risk of death.”. Conclusion at the end of the manuscript: “The ABSI serves as a reliable prognostic indicator in patients with severe burns. Unlike previous studies, this meta-analysis provides robust evidence for the clinical utility of the ABSI. Our findings reinforce that patients with elevated ABSI scores are at an increased risk of mortality, which emphasizes the importance of this indicator in risk stratification and in guiding treatment prioritization in modern burn care. However, due to the limitations of our meta-analysis and in the included studies, more high-quality studies are still needed to verify our findings.” Furthermore, about the comparison between ABSI and other scoring systems, we have addressed this issue in the limitation part: “Fourth, we did not compare the prognostic value of the ABSI with that of other similar indicators, such as the Baux score, which is also difficult to achieve in meta-analyses because of the lack of original data.”

Response to Journal editor:

Answer: We have revised the manuscript according to the PLOS ONE’s style requirements.

“This work was funded by the Key Projects of Science and Technology Department of Sichuan Province(2024YFFK0063).”

“This work was funded by the Key Projects of Science and Technology Department of Sichuan Province(2024YFFK0063).”

Answer: We have deleted the funding statement from the manuscript and provided corresponding information in the online submission. Cover letter was also updated.

Answer: The ORCID for the corresponding author has been updated.

Answer: The ethics statement has been moved to the Methods section.

5. We note that your Data Availability Statement is currently as follows: [All data used in this work have been presented in the manuscript.]

Answer: The Data Availability Statement has been updated as follows: “All data used in this work have been presented in the Supplementary Data File.” Meanwhile, the data used in this meta-analysis has been uploaded.

6. As required by our policy on Data Availability, please ensure your manuscript or supplementary information includes the following:

Answer: We have revised the manuscript as request. Notably, we uploaded a supplementary table listing the reasons for exclusion after reviewing the full texts.

Response to Reviewer #1:

This manuscript analyzes the prognostic relevance of the ABSI score based on 16 studies published between 2015 and 2024. The statistical analysis is thorough and the results are presented in a clear and understandable form. However, the overall quality of the text lacks in multiple areas.

Question 1: Overall, many of the chosen formulations are inadequate and articles are often missing, which makes the text not very pleasant to read.

Answer: Dear reviewer, thanks for your valuable comments. We have carefully checked and revised the manuscript thoroughly. Besides, this manuscript has been edited by the AJE service with the verification code 51D0-8C70-A160-CA8E-72A8.

Question 2: The authors claim, that the ABSI score could possibly be used as a novel prognostic score, which doesn't make any sense as the score was introduced in 1982 and has been in use ever since. It is true, that the relevance of the score has been questioned in the past, however, there are multiple (very recent) studies which test the score for its validity. However, the authors repeatedly claim in this manuscript that the ABSI score is novel, which could be very misleading for readers.

Answer: Dear reviewer, sorry for the worry description. We have carefully revised the manuscript. In the introduction section, we indicated that the ABSI was developed in 1982, cited the reference and also reintroduced the necessity of conducting this study as follows: “The abbreviated burn severity index (ABSI), which was introduced in 1982, is a prognostic indicator that is based on sex, age, inhalation injury, occurrence of third-degree burns and total burned area [14]. In recent decades, this tool has been investigated in several studies since 2015 and is believed to have some prognostic value in severely burned patients. However, the results of these studies varied, and the clinical value of the ABSI in predicting mortality in patients with severe burns still lacks high-level evidence. Therefore, for the first time, this meta-analysis aimed to further clarify the association between the ABSI and the risk of mortality in patients with severe burns”

Question 3: Furthermore, the authors don't provide enough references for the claims in the introduction as well as the discussion. Following claims need to be undermined with references:

Introduction:

- Young children and the elderly, due to thinner skin and weaker immune function, tend to have poorer outcomes when faced with burns of the same severity. (??)

- Additionally, underlying conditions such as diabetes, heart disease, and renal failure can impair the body's ability to recover from burns, leading to a poorer prognosis. (e.g. Resch A, Neumueller A, Christ A, Staud C, Hacker S. Chronic kidney disease and cardiovascular disease reduce survival rates after burn injury: A retrospective study over 20 years.)

- Severe burn patients are prone to infections, especially with extensive burns, and infection is one of the leading causes of death in these cases.

- However, due to weaker immunity and lower treatment compliance, children may have a worse prognosis compared to adult patients in the same situation. (The ABSI score is especially relevant for severely burned patients, who require intensive care treatment. There is no such thing as "lower treatment compliance" in children in ICU's.)

Answer: Dear reviewer, many thanks for your valuable comments. We have carefully added corresponding references to support these claims. Besides, this sentence “However, due to weaker immunity and lower treatment compliance, children may have a worse prognosis compared to adult patients in the same situation. (The ABSI score is especially relevant for severely burned patients, who require intensive care treatment” has been deleted.

Question 4: In the materials and methods section the authors write "Studies met foll

---

## [Decision Letter · Decision Letter 1]

29 Jan 2025

Association of the abbreviated burn severity index with mortality in severely burned patients: a meta-analysis

PONE-D-24-45843R1

Dear Dr. Li,

We’re pleased to inform you that your manuscript has been judged scientifically suitable for publication and will be formally accepted for publication once it meets all outstanding technical requirements.

Kind regards,

David M. Burmeister, PhD

Academic Editor

PLOS ONE

Additional Editor Comments (optional):

Reviewers' comments:

Reviewer's Responses to Questions

**Comments to the Author**

1. If the authors have adequately addressed your comments raised in a previous round of review and you feel that this manuscript is now acceptable for publication, you may indicate that here to bypass the “Comments to the Author” section, enter your conflict of interest statement in the “Confidential to Editor” section, and submit your "Accept" recommendation.

Reviewer #4: All comments have been addressed

2. Is the manuscript technically sound, and do the data support the conclusions?

Reviewer #4: Yes

3. Has the statistical analysis been performed appropriately and rigorously? 

Reviewer #4: Yes

4. Have the authors made all data underlying the findings in their manuscript fully available?

Reviewer #4: Yes

5. Is the manuscript presented in an intelligible fashion and written in standard English?

Reviewer #4: Yes

6. Review Comments to the Author

Reviewer #4: (No Response)

7. PLOS authors have the option to publish the peer review history of their article (what does this mean? ). If published, this will include your full peer review and any attached files.

**Do you want your identity to be public for this peer review?** For information about this choice, including consent withdrawal, please see our Privacy Policy .

Reviewer #4: No

---

## [Editor Report · Acceptance letter]

PONE-D-24-45843R1

PLOS ONE

Dear Dr. Li,

I'm pleased to inform you that your manuscript has been deemed suitable for publication in PLOS ONE. Congratulations! Your manuscript is now being handed over to our production team.

Kind regards,

on behalf of

Dr. David M. Burmeister

Academic Editor

PLOS ONE